Automated analysis of invadopodia dynamics in live cells

Berginski Matthew E. 1 7
Creed Sarah J. 2 8
Cochran Shelly 1
Roadcap David W. 2
Bear James E. 2 3 4 jbear@email.unc.edu
Gomez Shawn M. 1 5 6 smgomez@unc.edu
1 UNC/NCSU Joint Department of Biomedical Engineering, University of North Carolina at Chapel Hill , Chapel Hill, NC , USA
2 Department of Cell Biology and Physiology, University of North Carolina at Chapel Hill , Chapel Hill, NC , USA
3 Lineberger Comprehensive Cancer Center, University of North Carolina at Chapel Hill , Chapel Hill, NC , USA
4 Howard Hughes Medical Institute , Chevy Chase, MD , USA
5 Department of Computer Science, University of North Carolina at Chapel Hill , Chapel Hill, NC , USA
6 Department of Pharmacology, University of North Carolina at Chapel Hill , Chapel Hill, NC , USA
Perez-Acle Tomas
7 Current affiliation: Department of Biomedical Engineering, Duke University, Durham, NC, USA

8 Current affiliation: Monash Institute of Pharmaceutical Sciences, Monash University, Parkville, VIC, Australia

Electronic publication date: 2014 Jul 1
Publication date: 2014
Volume: 2
Electronic Location ID: e462
Received 2014 Apr 1; Accepted 2014 Jun 9
Copyright: © 2014 Berginski et al.
Copyright year: 2014
Copyright holder: Berginski et al.
License: This is an open access article distributed under the terms of the Creative Commons Attribution License, which permits unrestricted use, distribution, reproduction and adaptation in any medium and for any purpose provided that it is properly attributed. For attribution, the original author(s), title, publication source (PeerJ) and either DOI or URL of the article must be cited.
License URL: https://creativecommons.org/licenses/by/4.0/

Keywords: Invadopodia, Podosomes, Image analysis, Live cell imaging, Cancer, Fluorescence microscopy, Metastasis, ECM degradation, Invasion

Funding: HHMI NIH GM083035 UNC UCRF We gratefully acknowledge support from HHMI and NIH grant GM083035 to JEB, as well as support from the UNC UCRF to JEB and SMG. The funders had no role in study design, data collection and analysis, decision to publish, or preparation of the manuscript.

==============================
Multiple cell types form specialized protein complexes that are used by the cell to actively degrade the surrounding extracellular matrix. These structures are called podosomes or invadopodia and collectively referred to as invadosomes. Due to their potential importance in both healthy physiology as well as in pathological conditions such as cancer, the characterization of these structures has been of increasing interest. Following early descriptions of invadopodia, assays were developed which labelled the matrix underneath metastatic cancer cells allowing for the assessment of invadopodia activity in motile cells. However, characterization of invadopodia using these methods has traditionally been done manually with time-consuming and potentially biased quantification methods, limiting the number of experiments and the quantity of data that can be analysed. We have developed a system to automate the segmentation, tracking and quantification of invadopodia in time-lapse fluorescence image sets at both the single invadopodia level and whole cell level. We rigorously tested the ability of the method to detect changes in invadopodia formation and dynamics through the use of well-characterized small molecule inhibitors, with known effects on invadopodia. Our results demonstrate the ability of this analysis method to quantify changes in invadopodia formation from live cell imaging data in a high throughput, automated manner.

Introduction

Migration through a three dimensional environment, such as during embryonic development or metastasis, is a multistage process beginning with the migration of either single cells or groups of cells away from the primary site and into the surrounding ECM (Extra Cellular Matrix). To accomplish this migration, the ECM is commonly degraded, typically through the use of matrix metalloproteinases (MMPs) to form paths through which the cells can move. Invadopodia and podosomes, collectively termed invadosomes, are the two structures most commonly associated with this behavior. In the case of cancer cells and the process of metastasis, the formation of invadopodia that actively degrade the ECM is a common observation (Destaing et al., 2011; Murphy & Courtneidge, 2011). Importantly, while markers such as cortactin and Arp2/3 subunits can help discern these structures from other actin-rich structures, the direct observation of degradation activity is required in order to accurately identify these invasive structures.

Invadopodia were first imaged in Rous sarcoma virus transformed cells using several imaging methodologies including interference reflection and fixed cell labelling (Chen, 1989; Tarone et al., 1985). To measure degradation activity in vitro, quantitative imaging assays of invadopodia behavior have been developed that use fluorescently labeled ECM to visualize regions of degradation caused by invadopodia (Artym, Yamada & Mueller, 2009). By combining fluorescently labeled ECM and fluorescently labelled intracellular markers of invadopodia, such as cortactin (Artym et al., 2006), cofilin (Stoletov et al., 2013) or actin (Albiges-Rizo et al., 2009), the activity of single invadopodia can be followed through time (Sharma, Entenberg & Condeelis, 2013a). The continued development of methods to observe invadopodia formation and ECM degradation has made it possible to quantify the effect of siRNA knockdown or drug treatment on invadopodia dynamics (Beaty et al., 2013; Sharma et al., 2013b). However, the time-lapse image sets produced using these methods have traditionally been analysed using manual selection/analysis methods that are time-consuming and potentially biased in the selection of which invadopodia will be measured.

In order to improve the reliability of invadopodia measurement systems, we have developed a system to automate the segmentation, tracking and quantification of invadopodia in time-lapse fluorescence image sets at both the single invadopodia level and whole cell level. These methods use the fluorescently labeled ECM images to determine when and where individual invadopodia or cells are degrading the matrix, making detailed studies of invadopodia formation, timing and activity possible. Since these methods were initially developed using LifeAct-GFP as marker of invadopodia, the system does not assume that all bright puncta will become invadopodia and degrade the ECM and instead uses the changes in the underlying ECM to classify puncta. The first set of tools has been designed to identify and track single invadopodia and the corresponding ECM, as might be studied using images taken at 60× magnification. The second set of tools does not attempt to track individual invadopodia and instead focuses on cell population images which could be taken at 20× magnification. We have made both sets of tools available as open source packages and made a web-based version of the single invadopodia analysis available.

Materials and Methods

Reagents

All chemicals and reagents were purchased from Sigma (St Louis, MO) unless otherwise stated. The BB94, Purvalanol A, FAK inhibitor II and PP2 compounds were all purchased from EMD BioSciences (La Jolla, CA). All drugs were soluble in DMSO, which was used as a control for all experiments. The BB94 and FAK inhibitor II were used at a final concentration of 5 µM, while Purvalanol A was used at 2 µM and PP2 was used at 10 µM. The Alexa Fluor 568 protein labeling kit used to label the gelatin matrix was purchased from Invitrogen (Eugene, OR), as were all cell culture reagents. Fugene 6 used for transfections was purchased from Roche Diagnostics (Indianapolis, IN). For dynamic visualization of actin, the LifeAct peptide (Riedl et al., 2008) was cloned into the previously described pLL 5.0 GFP Lenti-viral base vector (Wang & McNiven, 2012).

Cell lines and culture

Wild Type and LifeAct-GFP WM2664 cell lines and 293FT Hek cells were maintained in DMEM with 10% FBS and 1% PSG at 37 °C in a humidified atmosphere with 5% CO2. Lentiviral production was performed as previously described in Wang & McNiven (2012). Briefly, lentiviral expression plasmids were transfected into 293FT cells with the construct of interest using Fugene 6. After 24 h, media is changed and viral media is applied to 30–50% confluent WM2664 cells in a 6-well dish for 48 h. Fluorescent cells were FACS sorted for the 20–80% range of GFP positive cells.

Fluorescent gelatin matrix

To prepare ECM substrate for the invadopodia assays, 0.2% Type A gelatin from porcine skin was fluorescently labeled using the Alexa Fluor 568 protein labeling kit according to the manufacturer instructions. Culture vessels were coated at room temperature with thin layers of 20% poly-L-lysine for 20 min followed by 0.5% glutaraldehyde (Electron Microscopy Sciences, Hatfield, PA) for 15 min. An 8:1 ratio of unlabeled to labeled gelatin was prepared and incubated on culture vessels for 10 min. Following gelatin incubation a 5 mg/mL solution of NaBH4 (Fisher Scientific, Fair Lawn, NJ) was incubated for 15 min to quench any remaining glutaraldehyde. Between each level of coating, the dishes were washed three times with phosphate buffered saline (PBS).

Single invadopodia assay imaging parameters

LifeAct-GFP WM2664 cells were plated (± inhibitors) onto heated bioptech dishes (Bioptech Inc., Butler, PA) coated with Alexa Fluor 568 labeled gelatin, as described above. Cells were allowed to equilibrate on the microscope 1 h before imaging. Random fields of view were selected and time-lapse microscopy was performed on an Olympus IX-81 inverted microscope (60×, 1.3NA objective) with a Hamamatsu CCD camera (model c4742-80-12AG) and a Prior Lumen200Pro epifluorescence system (Olympus America, Center Valley, PA) on a heated stage with heated lid receiving a constant stream of humidified 5% CO2 to maintain cell viability. Images were captured in both the 488 and 568 channels every 5 min for 12 h using MetaMorph Imaging software (Molecular Devices, Sunnyvale, CA).

Cell population assay imaging parameters

LifeAct-GFP WM2664 cells (± inhibitors) were plated onto MatTek 35 mm glass bottom dishes (MatTek Corporation, Ashland, MA) coated with Alexa Flour 568 labeled gelatin, as described above. Cells were allowed to equilibrate in the incubation chamber of the microscope, heated to 37 °C with a humidified atmosphere of 5% CO2, for 1 h before imaging. Cells were imaged in the Olympus VivaView microscope with a 20×/0.75NA objective and morotized magnification changer set to 1×, using a Hamamatsu Orca R2 cooled CCD camera (Olympus America, Center Valley, PA). A template was used to select 25 predetermined positions in each dish and images in the 488 and 568 channels were captured every 30 min for 25 h.

Invadopodia analysis

Before analyzing, the fluorescent ECM images were photobleach corrected. To ensure that the degradation of ECM was not incorrectly identified as photobleaching, only pixels outside the cell bodies were considered when photobleaching correction was applied. Images were then flat-field corrected and the single cell time-lapse images for single cell assays were registered (the cell population movies did not require registration). In the ECM images used for the single invadopodia analysis, the average fluorescence outside the cell bodies was set to 1000 to allow the local difference values to be compared between ECM preparations. The LifeAct-GFP images for the single invadopodia analysis were not pre-processed. The LifeAct-GFP images for the population analysis were flat-field corrected. The analysis software uses a high-pass filter and threshold to identify regions of high actin concentration (potential invadopodia) that co-localize with underlying ECM degradation to identify active invadopodia and track them through time.

Results

The results section is divided into two parts, the first of which deals with the analysis and quantification of single Invadopodia in 60× images. The second portion of the results details the analysis and quantification of ECM degradation by whole cells.

Identification and tracking of LifeAct-GFP puncta in 60× images

The development of software and subsequent analysis of invadopodia was performed in multiple steps. The first step of analysis involved the development of methods to segment and track single actin aggregates, termed puncta, from images of LifeAct-GFP. After identifying actin puncta, a set of measurements were made from the fluorescent ECM images at each time point to classify the effect of the puncta on the underlying ECM. As only a portion of the population of LifeAct-GFP puncta degrade the matrix, puncta that degrade the matrix are classified as invadopodia. Additional collected properties such as degradation rates were then used in further analyses as described below.

To identify LifeAct-GFP labelled puncta in WM2664 cells (Fig. 1A), untreated and BB94 treated cells were imaged by time-lapse microscopy. In order to ensure that the automated system would have decision thresholds similar to that of experts doing a manual analysis, prior to automated analysis, images were assessed by three experienced independent observers who manually identified and segmented LifeAct-positive puncta. A final consensus segmentation was reached by majority vote from the individual manual segmentations. These invadopodia segmentations, determined by our observers were then used to test potential segmentation strategies, threshold settings and determine appropriate filters.

Figure 1 Segmentation of puncta from LifeAct-GFP images.

(A) Epifluorescence image of WM2664 cell expressing LifeAct-GFP. (B) Image from Part A passed through a high-pass filter. (C) Contour plots showing the detection errors for identification of puncta seeds. (D) Locations of puncta seeds accepted with minimum seed size 6 and high-passed threshold of 3. (E) Error rates on a pixel basis as a function of the puncta expansion threshold. (F) Puncta area plotted versus ratio between major and minor axes in puncta manually identified in either control or BB94 treated cells. (G) Locations of segmented LifeAct-GFP puncta based on seeding, expansion, area and major over minor axes filtering.

The first stage of the automated segmentation pipeline used a high-pass filter to remove the background noise from the LifeAct-GFP signal (Fig. 1B) and then determined the mean and standard deviation of the high-pass filtered pixel intensities for use as thresholds (Haier et al., 2003). To identify individual puncta, a seed-based region-growing segmentation method was used. To identify seed pixels in each image, intensity thresholds from 1 to 10 standard deviations were tested and these automatically segmented regions were compared to those identified through manual segmentation (Fig. 1C), with the false positive and negative rates computed as previously described (Matov et al., 2010). As the seed threshold increased, the rate of false positives decreased, while the rate of false negatives increased. We also tested minimum seed sizes, ranging from 2 to 10 pixels, and observed the same general behavior in the false positive and false negative rates as seed size increased. To balance these factors, we empirically selected a standard deviation threshold of 3 and a minimum size of 6 pixels to identify the puncta seeds (Fig. 1D).

After identification of the puncta seeds, a second threshold was selected to expand around each of the identified seed regions. To assess the performance of the seed expansion procedure, we measured the degree of overlap between the manually segmented puncta and the matching computer segmented puncta. We tested thresholds from 0 to 3 standard deviations from the mean (Fig. 1E). As expected, as the seed expansion threshold increases, the false positive rate decreases, while the false negative rate increases. We selected a seed expansion threshold of 1.75 to balance these two factors. We note that thresholds may need to adjusted depending on the fluorescent marker or cell line used. As a final filter, we also considered the area and ratio between the major and minor axes of the segmented puncta. To determine these filters, we measured these properties in the manually identified puncta (Fig. 1F). We used the minimum and maximum values for the area and the major over minus axes ratio as filters for any objects identified after seed identification and expansion. The cell edge was also identified in the LifeAct-GFP signal using a previously published method (Hoshino, Branch & Weaver, 2013). The properties for each identified puncta (Fig. 1G) were then collected, which included area and the distance of the puncta from the nearest cell edge.

Each puncta was also tracked through the experiment, using overlap in adjacent frames to connect the segmented puncta. The majority of the identified puncta were present for only one frame (Fig. 2A), but a population of puncta that could be followed for 12 frames or more (Fig. 2A inset) was also observed. For puncta that live for 12 frames or more, the average area (Fig. 2B) and the average distance from the nearest cell edge (Fig. 2C) were calculated over time.

Figure 2 Properties of segmented and tracked puncta in control cells.

(A) Histogram of the lifetime of the segmented puncta. Inset graph shows the lifetime of puncta with lifetime of 12 frames or more. (B) Histogram of the average puncta area for puncta with lifetimes over 1 h. (C) Histogram of the distance to the nearest cell edge.

Determination of ECM degradation by puncta

As described previously, degradation of the matrix is a necessary condition for an individual puncta to be classified as a functional invadopodia. This degradation is visible as dark regions that develop in a fluorescently labelled gelatin matrix underneath puncta over time. In order to detect this change in the ECM, the average ECM intensity immediately underneath each puncta and the area in a five-pixel border surrounding the puncta was tracked over time (Fig. 3A). Areas within the surrounding border occupied by another identified puncta were excluded from quantitation. The difference between the average intensity in the surrounding ECM and the ECM underneath each puncta was used to calculate the local fluorescence difference, so that puncta that have degraded the ECM will have positive values in the local fluorescence difference, while non-degrading puncta will have values near zero. To account for irregularities in the gelatin matrix, the intensity of the matrix before the puncta appeared in the time-lapse image set was also determined. This pre-birth local fluorescence difference was calculated over the same pixels using the image immediately before the appearance of puncta. In cases where the puncta is present at the beginning of the time-lapse, the first image of the ECM time-lapse was used. The pre-birth local difference calculated at each image was used to correct the observed local intensity difference, giving the corrected local intensity difference.

Figure 3 Measurement of ECM degradation underneath single puncta.

(A) Cartoon representation of a single puncta and the corresponding ECM underneath that degrading puncta. (B) Small multiple visualization of a single degrading puncta from a control cell and corresponding ECM intensities. The puncta is outlined in green, while the region classified as the local background is shaded purple. The first column shows the LifeAct and ECM images immediately before puncta formation and the last column shows the same areas immediately after puncta disappearance. (C) Small multiple visualization of a non-degrading puncta. (D) Boxplots of the mean local corrected difference in four of the experimental conditions. The box indicates the 25th and 75th percentiles, while the bold line indicates the median and the whiskers extend to 1.5 times the interquartile range. * indicates p < 0.05 by t-test.

To classify the puncta as active invadopodia, the values of the local difference, pre-birth local difference and the corrected local difference were analysed. To ensure that sufficient data from each puncta was assessed, we limited our search to only those puncta present in the time-lapses for at least 1 h (12 images under our experimental protocol). Application of this filtering constraint left 2,323 untreated puncta and 979 DMSO, 294 BB94, 533 focal adhesion kinase (FAK) inhibitor, 83 PP2 and 125 Purvalanol A treated puncta. For invadopodia, we would expect the local difference and corrected local difference values to be positive (puncta that do not degrade the matrix; see a sample invadopodia in Fig. 3B) and for both of these values to average around zero or negative for non-invadopodia puncta (see a sample non-invadopodia puncta in Fig. 3C). Next, we tested whether the mean local intensity difference and the mean corrected local intensity difference were statistically different from zero using a t-test. After applying a Bonferroni correction for the number of tests run, 336 untreated, 44 control, 12 BB94 treated and 35 FAK inhibitor treated puncta were classified as invadopodia. As expected, no actively degrading invadopodia were detected in the PP2 or Purvalanol A treated cells. The mean local corrected difference was greater in the untreated, control and FAK inhibitor treated cells as compared to the BB94 treated cells (Fig. 3D). The calculated differences in invadopodia number following drug treatment correlate with previously identified outcomes on invadopodia for each inhibitor tested (Chan, Cortesio & Huttenlocher, 2009; Hoshino et al., 2012; Wang et al., 1994) and demonstrate the accuracy of the automated detection methods developed here. The 7 invadopodia identified in BB94 treated cells were manually identified as false positives and we excluded them from further analysis. The low number of false positives identified in the BB94 treated cells further demonstrates the improved performance capable through the use of these automated imaging methods.

Measurement of invadopodia properties

Invadopodia identified in the untreated, control and FAK treated invadopodia were used to measure the mean area of each invadopodia and average distance of invadopodia to the nearest cell edge (Figs. 4A and 4B). No differences were detected in the average area of invadopodia between conditions (Fig. 4A). The average distance of invadopodia from the edge of the cell was decreased by 28% in control cells compared to untreated cells and by 38% after treatment with FAK inhibitor (Fig. 4B). There was no difference observed in invadopodia distance from the edge between control cells and FAK inhibitor treated cells (Fig. 4B). The lifetime of invadopodia showed increases of 45% as compared to the untreated cells following treatment with the FAK inhibitor (Fig. 4C).

Figure 4 Properties automatically extracted from the identified invadopodia.

(A) The average area of invadopodia. (B) The average distance from the cell edge of invadopodia. (C) The lifetime of invadopodia. (D) The average time to maximum matrix degradation.

The amount of time from puncta formation until maximum degradation was also quantified. To measure this property a smoothed curve was fit to the degradation curve of each invadopodia (e.g., the curve shown in Fig. 3B). The earliest time point at which the smoothed values hit 90% of the maximum was chosen as the time to maximum degradation. The time to reach maximum degradation was increased by 77% in the FAK treated cells when compared with the untreated cells (Fig. 4D).

Quantification of whole-cell degradation behaviors

To complement the single invadopodia analysis, a system for quantifying the ECM degradation capacity of entire cell populations was also developed. As with the individual invadopodia work, WM2664 cells expressing LifeAct-GFP and exposed to the same drug treatments used for single cell analysis were used to test the capabilities of this system. After image pre-processing, cells in the LifeAct-GFP images (Fig. 5A) were identified using the same algorithm as that used to find the cell outline in the single invadopodia analysis (Fig. 5B). To minimize the number of cell clusters found and make sure results describe measurements for individual cells, we quantified the areas of all observed single cells and multi-cell clusters (Fig. 5C). We empirically set a minimum size threshold of 1,500 pixels (107 µm2) to exclude small debris and a maximum size threshold of 20,000 pixels (1422 µm2) to exclude cell clusters from further analysis.

Figure 5 Identification of LifeAct-GFP expressing WM2664 cells.

(A) Sample LifeAct-GFP image from a 25 h time-lapse experiment after photobleaching and flat-field correction. (B) Same image as in part A, with the segmented cells outlined in yellow. (C) The distribution of object sizes detected based on intensity thresholding. (D) The distribution of the lifetime of objects detected in the control time-lapse image sets. (E) Sample fluorescent ECM image from the same time and field as in part A. (F) The ECM channel in the same field as in part E, at the end of the 25 h experiment. The two sub-images show the first and last image of two cells outlined in red identified in the field. The lower portion of each sub-image shows the ECM channel immediately underneath each cell.

By identifying and tracking individual cells through time, we observed a bimodal distribution of cell lifetimes (Fig. 5D). Objects present for less than 10 h were often non-adhered cell bodies or other debris in the field of view and were excluded from further analysis. In parallel with the cell images, images were also taken of the fluorescent ECM (Fig. 5E). Using the segmented cells and the intensity of the underlying matrix, we were able to observe both cells that degraded the matrix as well as those that did not (Fig. 5F).

Determination of cellular ECM degradation by whole cells

The segmented cell and ECM images (Fig. 5) were used to classify cells as either degraders or non-degraders. To accomplish this, methods similar to the analysis of single invadopodia were used (Fig. 6A). More specifically, for cells identified in each image, the average ECM intensity underneath and within a 40 pixel border around each cell was measured, while excluding any region overlapping with another cell and the corresponding regions in the prior ECM image. The average change in fluorescence intensity from the prior image to current image and in the surrounding 40 pixel border was then calculated. To allow these values to be compared across differing ECM intensity regions, the values were saved as the percentage difference between the ECM intensity underneath each cell and the surrounding region. Therefore, invading cells (i.e., those degrading the matrix) would be expected to have lower ECM intensities immediately underneath the cell. We then collected time-series images for each cell present for at least 10 h (example time-series is shown in Fig. 6B).

Figure 6 Determination of single cell degrader status.

(A) Cartoon representation of two cells of which one cell degrades the matrix (Cell #1) and another non-degrading cell (Cell #2). Also shown are the overlapping areas of the cell location and the results of comparing the first and last images from the image set. (B) Example small multiple tracks of single cells through time and the corresponding measurement of the percent of matrix degraded between each image. The colors outlining each image on the left corresponds to the same color line in the plot on the right. (C) Boxplots of the overall percentage of fluorescent ECM removed underneath control, DMSO treated and BB94 treated cell. * indicate p < 0.05 via t-test.

In addition to the image-by-image assessment of the percentage of the ECM degraded, the overall percentage of fluorescent ECM degraded by each cell was also assessed (Fig. 6C). To find the overall degradation percentage for each cell, the area of influence for each cell was determined by finding the amount of time the cell covered each pixel location in the field of view. Any pixel location covered for at least 2.5 h was considered to be in that cell’s area of influence. The change in fluorescence intensity from the first image of the ECM time-lapse and the area surrounding the cell was then calculated in the same manner as when calculating the percentage change from image to image.

Each cell was classified as degrading or non-degrading on a per image basis using the between images and total ECM degradation percentages calculated. As the BB94-treatment is expected to block all activity of the MMPs, degradation percentages found in the BB94 treated cells were used as a negative control. We empirically assessed the potential cut-off values to minimize the number of false positives (i.e., the number of BB94-treated cells classified as degraders).

Measurement of matrix degrading behaviors in whole cells

With each cell classified as a degrader or non-degrader, we calculated the percentage of cells in each class at half hour intervals throughout the experiment (Fig. 7A). In general, treatment with the FAK inhibitor tended to increase the percentage of degrading cells, while Purvalanol A decreased the percentage of degrading cells. Only eight degrading cells were detected under BB94 treatment and these were manually conformed as false positives and excluded from further analysis. These results are comparable to those observed at the single invadopodia level and again confirm the accuracy and low rate of false positives obtained using the developed automated analysis methods.

Figure 7 Properties of degrading cells.

(A) Percentage of cells classified as degraders over the 25 h imaging time frame. Percentages are averages from several experiments (n, 5 control; n, 2 BB94; n, 14 DMSO; n, 4 PP2; n, 3 Purvalanol A and n, 4 FAK inhibitor). (B) The total area and (C) rate of degradation from each cell classified as an invader, ∗p < 0.05 via t-test.

The area of ECM degraded by each cell was also quantified. To find the degraded regions of the matrix, the first and last images of the time-lapse were compared; any region where the intensity had decreased by 20% was marked as degraded. These degraded areas were assigned to each cell according to the previously defined area of influence (Fig. 7B). FAK inhibitor significantly increased the area degraded by each cell (62% increase compared to control), while PP2 and Purvalanol A each decreased the average area degraded (61% decrease compared to control). Similarly, the rate of degradation was also quantified by dividing the total area degraded by the cell lifetime (Fig. 7C). For this property we observed similar trends to the total area degraded with FAK inhibitor treated cells having a higher rate of degradation (23% increase compared to control), while in the PP2-treated cells and Purvalanol A-treated cells, there was a decrease in the degradation rate (37% and 43% decreases compared to control).

Discussion

Understanding the process of metastasis is an as yet unresolved issue in cancer biology and whose study has important ramifications in disease management and therapy. As the escape and migration of cancer cells from the primary tumor is preceded by degradation of the ECM, invadopodia may play a highly significant role. However, the quantification of their dynamic behaviors has been relatively limited. The framework presented here provides a reliable approach for the quantitative analysis of invadopodia behavior in both single cells and in cell populations over time. To develop this system, we gathered time-lapse image sets of the WM2664 metastatic cancer cell line expressing LifeAct-GFP (Riedl et al., 2008) forming invadopodia over an Alexa 568-labeled ECM. Since F-actin, as labeled by LifeAct, is not a conclusive marker of invadopodia on its own, we used the images from the labeled ECM to classify each F-actin puncta as either an invadopodia or not based on changes in the ECM intensity over time. After classification of the puncta, our system calculates several dynamic invadopodia properties such as lifetime and the time taken to reach maximum degradation levels.

To complement the analysis conducted at the single invadopodia level, we also designed an automated system that follows and quantifies degradation activity at the whole cell level. This approach uses images taken at a lower magnification (20× in our experiments) to gather a representative picture of the degradation behavior of cellular populations through time. Using this system, we can begin to explore dynamic aspects of cancer cell heterogeneity at the single cell level. For instance, the percentage of cells that have degraded the matrix as well as the rate and total amount of degradation performed by each cell can be quantified. Both of these systems were tested using a set of small molecule inhibitors previously demonstrated to block or enhance invadopodia formation (Chan, Cortesio & Huttenlocher, 2009; Hoshino et al., 2012; Wang et al., 1994). Our results are supported by earlier findings but also significantly extend the amount and degree to which invadopodia and cell degradation behaviors can be quantified, all within an automated image analysis framework.

Many different pharmaceuticals and their potential effects on invadopodia have been examined in fixed or live cell assays in the past. These include drug treatments such as BB94, Purvalanol A, and PP2 which are hypothesized to inhibit the formation of invadopodia (Chan, Cortesio & Huttenlocher, 2009; Hoshino et al., 2012; Wang et al., 1994). BB94 inhibits MMPs which are the enzymatic components of invadopodia that degrade the matrix (Wang et al., 1994). The effect of BB94 treatment is that formed puncta do not go on to degrade the matrix and are thus not formally classified as invadopodia (observation of degradation is a necessary requirement for classification of puncta as invadopodia) (Linder, Wiesner & Himmel, 2011). Live cell imaging experiments of cancer cells treated with BB94 act as means to set thresholds to minimize the number of false positives detected by the system. Purvalanol A inhibits cyclin-dependent kinase which in turn inhibits the tyrosine kinase Src, and PP2 is a more direct inhibitor of the tyrosine kinase Src and other members of the Src family. Src is known to be involved in invadopodia formation that consequently decreases when Src is inhibited by either Purvalanol A or PP2 (Chan, Cortesio & Huttenlocher, 2009; Hoshino et al., 2012). On the other hand FAK inhibitor II is hypothesized to enhance the formation of invadopodia. Studies using FAK knockdown have demonstrated an increase in invadopodia suggesting FAK regulates and suppresses invadopodia formation (Chan, Cortesio & Huttenlocher, 2009). Therefore FAK inhibitor II, which decreases FAK autophosphorylation and activation, should enhance invadopodia formation. Automated analysis as presented in the current study has shown the same decrease in invadopodia formation using BB94, PP2 and Purvalanol A, as well as an increase in invadopodia formation following treatment with FAK inhibitor II. Our results clearly demonstrate that the software developed accurately detects known changes in invadopodia formation in response to characterized perturbations.

A wide range of other fluorescent probes and tools are also capable of being used in the automated analyses described here, such as Tks5 (Tang et al., 2013) or cortactin (Artym et al., 2006). Invadopodia proteins need to be fluorescently tagged and be present at invadopodia during the degradation process in order to be assessed using our system. Many of these alternative tags should, in fact, more reliably mark invadopodia than the LifeAct F-actin label used in this work, making analysis with these markers less likely to result in false positives and decrease the requirement for manual confirmation of potential false positives as required in the BB94 samples in the current study. We note that the invadopodia described here were very similar to the 20–60 min lifetimes characterized in a breast cancer cell line, though this may also depend on the marker used (Beaty et al., 2013; Sharma et al., 2013b). The cell population analysis system can also be adapted to use alternative cell markers such as dyes or membrane associated fluorescent markers and may be further improved by the addition of a nuclear marker, making it possible to reliably split cell clumps using a watershed segmentation (Malpica et al., 1997). Alternative ECM labeling methodologies such as dye-quenched gelatins, or other ECM substrates such as fibronectin can similarly be utilized. This analysis tool is readily adaptable outside of the field of cancer research, for instance, for the examination of the related invasive structure podosomes, which are found in highly migratory cells such as osteoclests and macrophages (Albiges-Rizo et al., 2009; Block et al., 2008).

The software to process the labeled puncta in single cells and in cell populations through time has been released as open source packages available through the Gomez lab GitHub repository (https://github.com/gomezlab/). In addition, the single cell analysis of individual puncta can be performed through a web application (http://ias.bme.unc.edu/), which does not require the user to download or install any software to process a set of invadopodia images and also allows the ability to adjust thresholds to appropriate values. These two complementary analysis systems allow the quantification of invadopodia behavior at the single invadopodia and single cell levels. Combined with high throughput imaging methodologies, this analysis tool will be highly useful in screening small molecule inhibitors for efficacy in inhibiting invadopodia formation in cancer cells as well as quantifying genetic and cellular heterogeneity that may underlie related metastatic behaviors. Abbreviations

DMSO dimethyl sulfoxide

ECM extracellular matrix

FAK focal adhesion kinase

PBS phosphate buffered saline

MMPs matrix metalloproteinases

We gratefully acknowledge the UNC-Olympus Imaging Research Center, UNC Research Computing, as well as members of the Bear and Gomez labs for critical discussions and comments on the manuscript.

Additional Information and Declarations

Competing Interests

Author Contributions

Data Deposition

James E. Bear is an employee of the Howard Hughes Medical Institute. Shawn M. Gomez is an Academic Editor for PeerJ.

Matthew E. Berginski and Sarah J. Creed conceived and designed the experiments, performed the experiments, analyzed the data, wrote the paper, prepared figures and/or tables, reviewed drafts of the paper.

Shelly Cochran performed the experiments.

David W. Roadcap conceived and designed the experiments.

James E. Bear and Shawn M. Gomez conceived and designed the experiments, wrote the paper, reviewed drafts of the paper.

The following information was supplied regarding the deposition of related data:

Code is available as an open source package at: https://github.com/gomezlab/single_invado_analysis

https://github.com/gomezlab/invado_population.

Single-cell analysis can also be performed through a web application at:

http://ias.bme.unc.edu.

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
