# Peer review of "Automated analysis of invadopodia dynamics in live cells"

_PeerJ, doi:10.7717/peerj.462_

## Round 0.1 · original submission · Minor Revisions

This is a well written and very interesting work on a topic not usually covered by using pattern matching and machine learning approaches. However, to be accepted for publication at PeerJ, there are some concerns raised by the reviewers (see bellow) that should be carefully addressed.

Reviewer 1 ·

Basic reporting

No Comments

Experimental design

No Comments

Validity of the findings

No Comments

Additional comments

Overall this is an excellent manuscript, well written and detailing a non-biased method much needed in the field. The sharing of the analysis software is a nice bonus as well. That said, I have two issues that should be addressed before publication:

1. While written as a methods manuscript, there are biologically-relevant results that have been generated. For instance, in Figure 2, the individual properties of single invadopodia are displayed, but the biological significance is not mentioned. How do the average invadopodia area, distance from the edge, etc in WM2664 cells compare to what has been reported in the field? Such mention would strengthen the validity of the automation.

2. Of greater concern is that all of the reported analysis has been conducted in the WM2664 line. While there is no doubt this line makes invadopodia, it is not a line common to the field. It would be of greater benefit and would increase the validation of the software if single invadopodia and whole-cell degradation analysis were conducted on one or two well characterized invadopodia-producing lines. Why not test the model on EGF-stimulated MTLn3 or MDA-MB-231 cells? Or with SCC61 cells, which spontaneously make invadopodia? Similar results using the author's software to what is published would better validate the model across different laboratories, which is a main goal of the study.

Reviewer 2 ·

Basic reporting

The article reports an automated method to study and quantify invasdopodia in time lapse fluorrescence images. Overall it meets the professional standards required. The results section would be better placed after meterial and methods instead of after the introduction, to meet the journal style, but this is something I consider a minor change. Only a few small errors could be find in the writing:
ECM in the introduction should be defined
In the abstract section, the first senctence would look better split into two (Multiple cell types from specilized ......that are collectivelly referred to as invadosomes. Invadosomes are used by.....).
the computational part (algorithm and paramter optimization) fits better in the methods section, even if as it is, the paper reads well.

Experimental design

With respect to technical details, as with the writting, the article properly explains the methodology used. There are some technical details that would need better and more detailed explanation:
It is no clear how the thresholds have been selected. Explicit separation between train and test data needs to be explained (train data to select the thresholds to then report performance on the test data). In classification problems the common approach for threshold selection is to select the breakeven point (Precission equal to Recal in a P-R curve).
An important point is the the method described uses low magnification images (20x), but this is only mentioned in the discussion section

Validity of the findings

See changes suggested in the Experimental Design comments

Additional comments

Overall the paper meets quality standards required but there are some points that need better explanation and the text needs some re-ordering for a better understanding.

---

## Round 0.2 · accepted · Accept

Thanks a lot for making a compelling effort to address reviewers comments and corrections. I personally consider that this paper exposes an interesting methodology to both detect and characterise invadopodia dynamics in cancer cells. Therefore, it should be of broad interest to PeerJ readers.